# Transcriptomic and Metatranscriptomic Analyses Provide New Insights into the Response of the Pea Aphid *Acyrthosiphon pisum* (Hemiptera: Aphididae) to Acetamiprid

**DOI:** 10.3390/insects15040274

**Published:** 2024-04-15

**Authors:** Zhiyan Cai, Xuhui Zhao, Yuxin Qian, Kun Zhang, Shigang Guo, Yunchao Kan, Yuqing Wang, Camilo Ayra-Pardo, Dandan Li

**Affiliations:** 1Henan Key Laboratory of Insect Biology in Funiu Mountain, Henan International Joint Laboratory of Insect Biology, College of Life Science and Agricultural Engineering, Nanyang Normal University, 1638 Wolong Road, Nanyang 473061, China; czy_rico@126.com (Z.C.); 17354620935@163.com (X.Z.); 15670233632@163.com (Y.Q.); ov9v24lap8w@126.com (K.Z.); 18751633595@139.com (S.G.); kanyunchao@163.com (Y.K.); 2School of Life Science and Technology, Henan Institute of Science and Technology, 90 East of Hualan Avenue, Xinxiang 453003, China; 3Scientific Research Center, Nanyang Medical College, Nanyang 473061, China; wyq2013@163.com; 4CIIMAR—Interdisciplinary Centre of Marine and Environmental Research, Terminal de Cruzeiros do Porto de Leixões, University of Porto, Avda. General Norton de Matos s/n, 4450-208 Matosinhos, Portugal

**Keywords:** omics, insecticide resistance, symbiotic bacteria, gene expression analysis, insect bioassays

## Abstract

**Simple Summary:**

Acetamiprid, commonly used in agriculture to control aphids, faces the problem of resistance development in certain aphid species. To investigate the insect mechanisms involved in the adaptation process to acetamiprid, we compared the gene expression and microbial profiles of a pea aphid strain selected with acetamiprid (RS) with those of a non-selected strain (SS) using advanced omics techniques. The overall analysis revealed significant changes in the expression of genes involved in carbon and fatty acid metabolism in RS compared to those of SS aphids. In particular, we found increased expression of specific genes related to the synthesis of the components of the epidermal wax layer, suggesting that adaptation to acetamiprid involves the synthesis of a thicker protective layer. Additionally, subtle shifts in the bacterial composition of RS were detected. These results contribute valuable insights into the mechanisms underlying the pea aphid’s response to acetamiprid exposure. Such understanding is essential for informing future research efforts and developing more effective strategies to control this pest.

**Abstract:**

Acetamiprid is a broad-spectrum neonicotinoid insecticide used in agriculture to control aphids. While recent studies have documented resistance to acetamiprid in several aphid species, the underlying mechanisms are still not fully understood. In this study, we analyzed the transcriptome and metatranscriptome of a laboratory strain of the pea aphid, *Acyrthosiphon pisum* (Harris, 1776), with reduced susceptibility to acetamiprid after nine generations of exposure to identify candidate genes and the microbiome involved in the adaptation process. Sequencing of the transcriptome of both selected (RS) and non-selected (SS) strains allowed the identification of 14,858 genes and 4938 new transcripts. Most of the differentially expressed genes were associated with catalytic activities and metabolic pathways involving carbon and fatty acids. Specifically, alcohol-forming fatty acyl-CoA reductase (FAR) and acyl-CoA synthetase (ACSF2), both involved in the synthesis of epidermal wax layer components, were significantly upregulated in RS, suggesting that adaptation to acetamiprid involves the synthesis of a thicker protective layer. Metatranscriptomic analyses revealed subtle shifts in the microbiome of RS. These results contribute to a deeper understanding of acetamiprid adaptation by the pea aphid and provide new insights for aphid control strategies.

## 1. Introduction

Aphids, small sap-sucking insects of the order Hemiptera, are globally recognized agricultural pests that are notorious for causing significant damage to crops through their feeding activity and virus transmission [1,2]. Their economic importance and the challenges they pose to farmers worldwide require effective control strategies to minimize their impact on crop health and yield. Acetamiprid, a neonicotinoid insecticide, is widely used to control aphids due to its systemic action and broad spectrum of activity [3]. However, concerns have been raised about the overuse of acetamiprid, its potential impact on human health and the environment, and the development of resistance in target pests [4].

At the center of the debate on aphid control is the phenomenon of insecticide resistance, a widespread problem that has become a significant obstacle to pest control [5,6]. Insecticide resistance arises from a complex interplay of genetic, physiological, and environmental factors, including the rapid evolution of resistance mechanisms within aphid populations. Resistance mechanisms include target site insensitivity, metabolic detoxification, and behavioral adaptations, which reduce the efficacy of chemical insecticides [7,8]. Recent studies have documented resistance to acetamiprid in several aphid species [9,10,11,12,13]. For example, in one study, acetamiprid resistance in the melon aphid *Aphis gossypii* (Glover, 1877) was linked to the overexpression of several P450 genes [13]. Suppression of their expression through RNA interference (RNAi) dramatically increased sensitivity to acetamiprid. However, in a laboratory strain of *Myzus persicae* (Sulzer, 1776), resistance to acetamiprid was not accompanied by increased activity of P450 enzymes or any other known resistance mechanism [14], suggesting that different species may develop resistance to the same insecticide through divergent mechanisms.

In recent years, research has increasingly focused on the role of symbiotic bacteria in shaping aphid physiology and mediating responses to insecticides [15,16]. Symbiotic bacteria play a central role in the growth and development of aphids [17]. The primary symbiont *Buchnera aphidicola*, for instance, ensures its ubiquitous presence in aphids through vertical transmission via the ovaries, maintaining its continuity from one generation to the next. *Buchnera* provides essential amino acids and vitamin B, important nutrients that are often limited in the plant phloem [18,19,20]. Another prevalent symbiont, *Wolbachia*, affects host reproduction to promote its propagation and transmission [21,22,23,24]. In addition, aphids can harbor secondary or facultative symbionts [25,26], which are transmitted both vertically and horizontally within and between species. *Rickettsiella*, a facultative endosymbiont, induces a change in the body color of aphids from red to green [27], potentially protecting them from natural enemies. *Arsenophonus nasoniae* influences the sex ratio of the offspring of parasitic wasps by selectively killing male embryos [28,29]. *Hamiltonella* strains that are efficiently transmitted to the aphid offspring confer robust protection against dominant parasitoid species [30]. Li et al. observed that infection with *H. defensa* decreased the sensitivity of aphids to insecticides, particularly at low concentrations [31].

The pea aphid *Acyrthosiphon pisum* (Harris, 1776) is known worldwide as an important pest of legumes and pulses [32,33], where it causes significant economic losses. Despite the economic impact, there are currently no effective non-chemical methods to control this pest, so farmers rely heavily on chemical insecticides for control. Although resistance of the pea aphid to insecticides has not been extensively documented, a recent study found a high level of phenotypic resistance to several pyrethroids in field-collected strains compared to a susceptible reference strain [34]. Biochemical and molecular analyses indicated the involvement of P450 and esterases in resistance: increased P450 and esterase activities were observed in the resistant strains, and transcriptome profiling identified the P450 gene *CYP6CY12* as highly overexpressed. These results highlight the occurrence of pyrethroid resistance in the pea aphid and the importance of understanding the molecular mechanisms underlying the development of resistance to other insecticides commonly used to control this species, including neonicotinoids.

In this study, we analyzed the transcriptome and metatranscriptome of pea aphids selected with acetamiprid for nine consecutive generations. Our aim was to investigate the constitutive changes in gene expression and microbiome of the acetamiprid-selected strain compared to the non-selected strain to identify possible target mechanisms and symbionts involved in the adaptation process.

## 2. Materials and Methods

### 2.1. Aphids Rearing, Acetamiprid Exposure and Toxicity Assays

Aphids were reared according to the method described by Chang et al. [35]. A single apterous viviparous parthenogenetic *A. pisum* female was reared on broad bean (*Vicia faba* L.) seedlings in an incubator with a 16:8 h light–dark cycle, 20 °C temperature, and 60% relative humidity (RH).

For acetamiprid exposure, third-instar aphid nymphs were placed on leaves previously dipped in a 6.25 μg/mL acetamiprid solution in 0.01% dimethyl sulfoxide (DMSO) for 15~20 s, as described in Li et al. [31]. Acetamiprid acts as a systemic insecticide, controlling target insects through both contact and ingestion. Surviving aphids were transferred to fresh leaves after 72 h. This procedure was replicated in three independent experiments with 10 individuals in each replication. Aphids on leaves dipped in 0.01% DMSO served as a negative control and were designated as the non-selected (SS) strain. Acetamiprid exposure continued for nine consecutive generations to establish an acetamiprid-selected strain (RS). The criterion for selecting the RS strain at the ninth generation was based on observed differences in the growth cycle compared to the SS strain, with the RS strain requiring an additional 24 h to reach the adult stage.

Toxicity bioassays were performed as per standard leaf dip toxicity bioassay, with minor modifications [36]. Briefly, serial dilutions of an acetamiprid stock in 0.01% DMSO were prepared with 0.1% Triton X-100 in water. Medium-sized broad bean leaves were dipped into acetamiprid dilutions (12.5 μg/mL, 6.25 μg/mL, 3.12 μg/mL, 1.56 μg/mL, 0.78 μg/mL) for 30 s each and then laid flat on a non-absorbent plastic to air dry for one hour. Control leaves were treated with 0.01% DMSO in 0.1% Triton X-100 alone. Thirty pea aphids were exposed to each concentration. Assay plates were incubated with a 16:8 h light–dark cycle, 20 °C temperature, and 60% RH, and mortality was recorded after three days (i.e., dead aphid failed to respond after gentle prodding). The bioassays were repeated three times.

### 2.2. Transcriptomic Sequencing and Analyses

Total RNA of the third-instar nymphs from RS and SS was extracted using the TRIzol (Thermo Fisher, Waltham, MA, USA) method. Three independent experiments were conducted, and each sample had 30 individuals. Transcriptome libraries were constructed according to the method described by Wu et al. [37]. Raw reads were processed to remove 3′-adaptors and repeating reads. Clean reads underwent de novo assembly using the Trinity (version 2.0.6), TGICLL (version 2.1), and Phrap (Release 23.0) programs. The library was sequenced using the DNBSEQ (PE150, BGI, Beijing, China) according to the manufacturer’s instructions.

Clean reads were aligned to the NCBI non-redundant (NR) protein database, Swiss-Prot, Kyoto Encyclopedia of Genesa and Genomes (KEGG), and Cluster of Orthologous Group (COG) databases using Blastx (E-value ≤ 1 × 10^−5^). Unigene sequences were aligned to protein databases (NR, Swiss-Prot, KEGG, and COG). Blast2GO (ver. 2.5.0) was used for gene ontology (GO) annotation of unigenes with the NR database [38]. WEGO 2.0 software [39] was then used to perform functional classification of the GO terms for all unigenes. Pathway assignments followed the KEGG database. Unigene expression calculations used the FPKM (RPKM) method [40,41].

### 2.3. Metatranscriptomic Sequencing and Analyses

The total RNA of the third-instar nymphs from RS and SS was extracted using the TRIzol method. Three independent experiments were conducted, and each sample had 30 individuals. After rRNA removal, fragments underwent end repair and subsequent 3′ adenylation, followed by ligation of adaptors to the 3′ adenylated ends. The clean reads were obtained using SOAPnuke (version 1.5.0), Bowtie2 (version 2.2.5), and Samtools (version 1.2). High-quality reads were de novo assembled using MEGAHIT v1.2.9 software [42]. The qualified library was sequenced using DNBSEQ (PE100, BGI, Beijing, China) according to the manufacturer’s instructions. The raw data from sequencing were used for subsequent bioinformatics analyses.

To generate annotation information, the protein sequences of genes were aligned against KEGG, COG, and Swiss-Prot databases using DIAMOND (E-value ≤ 1 × 10^−5^) [43]. In contrast to the transcriptome, the taxonomic annotation relied on the Kraken LCA algorithm [44]. Based on the abundance profiles of species, the features (Genera, Phyla, and KOs) with significantly differential abundances between groups were determined using ANOVA. Differentially enriched KEGG pathways were identified [45,46].

### 2.4. Quantitative Real-Time PCR (qRT-PCR)

The total RNA of the third-instar nymphs from RS and SS was extracted with the TRIzol method. The first strand of cDNA was synthesized with 2 μg total RNA by PrimeScript II 1st Strand cDNA Synthesis Kit (TaKaRa, Dalian, China) using oligo d(T)15. qRT-PCR was performed according to the method described by Chang et al. [35]. The PCR program consisted of 40 cycles of denaturation at 95 °C for 30s, annealing at 55 °C for 30 s, and extension at 72 °C for 30s. The primers used are listed in Table 1. The 2^−∆∆Ct^ method was used to calculate the relative expression of mRNAs from the Cts obtained in the PCR quantification (Ct is the cycle threshold, which indicates the number of cycles experienced when the fluorescent signal in each reaction tube reaches a set threshold). ΔCt represents the average Ct value of the sample minus the internal control. ΔCtΔCt represents the average Ct value of the sample minus the control sample. Three independent experiments were conducted, and each sample was repeated three times. Three independent experiments were conducted (with 30 individuals for each), and each sample was repeated three times.

### 2.5. Data Analysis

The data were processed using SPSS Statistics 22.0 software. One-way analysis of variance (ANOVA) in conjunction with Tukey’s post-test and Student’s t test, both at *p* < 0.05 as the significance level, were performed to determine differences between treatments. Three replicates were performed for each treatment, and similar results were obtained. The standard error of the means was used to compare replicates.

Bioassays were analyzed using the open-source R environment [47]. Estimates of LC_50_, LC_99,_ LC_5_, 95% fiducial limits, and slopes were calculated by maximum likelihood logit regression analysis in a generalized linear model from individually fitted analyses of deviance as previously described [48]. Pairwise comparisons of LC_50_ values were significant (*p* < 0.05) when their respective 95% fiducial limits did not overlap [49].

## 3. Results

### 3.1. Acetamiprid-Resistant Aphids

Insect bioassays showed that after nine consecutive generations of exposure to the insecticide, RS exhibited a significant 1.9-fold reduction in susceptibility to acetamiprid compared to SS (Table 2). Although RS displayed notably higher LC_50_ and LC_99_ values than SS, there were no apparent differences in LC_5_ values between the two strains (*p* > 0.05; overlapped 95% fiducial limits). This indicates similar susceptibility at lower, sublethal acetamiprid concentrations.

### 3.2. Transcriptomic Profiling Associated with Acetamiprid Adaptation

RNA libraries were constructed and sequenced for RS and SS. This generated 45.57 and 46.16 million raw reads, respectively. After the removal of low-quality reads and adaptor sequences, 43.13 and 42.42 million clean reads were obtained from RS and SS, respectively (Appendix A). The analysis identified 14,858 expressed genes and 4938 new transcripts [50].

Expression analysis revealed 581 upregulated and 1220 downregulated genes in RS (Figure 1A). Among them, 15 genes showed significant upregulation, while 92 genes displayed significant downregulation in RS (*p* < 0.01) (Figure 1B). Upregulated genes included V-type H+-transporting ATPase subunit B (ATPeV1B), alanyl aminopeptidase (ANPEP), alcohol-forming fatty acyl-CoA reductase (FAR), while downregulated genes included alpha-N-acetylgalactosaminidase (NAGA), SRA stem-loop-interacting RNA-binding protein, mitochondrial (SLIRP), and polyhomeotic-like protein 1 (PHC1), among others (Table 3).

qRT-PCR experiments were performed to further validate the differentially expressed genes (Figure 1C). In general, the qRT-PCR results were consistent with the RNA-Seq results, except for ANPEP, which was significantly downregulated. Genes that were significantly upregulated in RS (*p* < 0.01) include FAR and ACSF2. FAR catalyzes the formation of fatty alcohols from fatty acids and is a key enzyme involved in the synthesis of the epidermal wax layer. ACSF2 catalyzes the formation of fatty acyl-CoA from fatty acids and thus promotes the utilization of these fatty acids.

Gene Ontology (GO) analysis revealed that the majority of the differentially expressed genes (DEGs), totaling 628 genes (Figure 2A), were primarily associated with catalytic activity. These genes were mainly implicated in cofactor biosynthesis and the metabolism of carbon and fatty acids. Protein network analysis of these 628 genes identified three key nodes: triosephosphate isomerase (TPI), glucose-6-phosphate isomerase (PHI), and delta-1-pyrroline-5-carboxylate synthetase (P5CS). Notably, the expression of these key nodes decreased in the RS strain (Figure 2B). TPI and PHI serve as key enzymes in glycolysis, while P5CS plays a critical role in proline metabolism, suggesting metabolic reprogramming to cope with stress in the RS strain.

The Kyoto Encyclopedia of Genes and Genomes (KEGG) pathway database assigned the differentially expressed genes to cofactor biosynthesis, carbon and fatty acid metabolism, and the ribosome pathways (Figure 2C). The expression of most genes assigned to metabolic pathways decreased in the RS strain (Figure 2D,E). In the ribosome pathway, however, gene expression increased in the RS strain, with the sole exception of the gene encoding the large subunit of ribosomal protein LP1 (RPLP1) (Figure 2F). GO and KEGG analysis showed that most DEGs in RS were concentrated in metabolic pathways and experienced partial repression. However, the induction of genes in the ribosome pathway suggests the activation of a potential adaptive response to the stress associated with acetamiprid exposure.

### 3.3. Metatranscriptomic Profiling Associated with Acetamiprid Adaptation

Metatranscriptomic sequencing was used to analyze the differences in bacterial abundance between RS and SS. A substantial dataset was generated, yielding 240.40 and 240.39 million clean reads, with 24.04 billion and 24.01 billion clean bases from RS and SS, respectively (Appendix A). Species annotation revealed a discrete increment in the abundance of four species, i.e., *B. aphidicola*, *Herbasepirillum huttiense*, *Deelftia acidovorans*, and *Lactobacillus iners*. Conversely, three species—*Serratia symbiotica*, *Escherichia coli,* and *Acinetobacter soli*—exhibited decreased abundance (Figure 3A and Table 4).

Expression analysis unveiled 24 significantly upregulated genes and 29 significantly downregulated genes (*p* < 0.05) (Figure 3B). GO annotations indicated that most downregulated genes were related to actin filament organization, regulation of actin filament organization, and cell–substrate adhesion (Figure 3C). The actin cytoskeleton, an intracellular structure involved in the onset and control of cell shape and function, was found to regulate the ion channel activity (Figure 3D).

The sequencing results were further validated by qRT-PCR. The results showed significant (*p* < 0.05) downregulation of genes encoding MAP7, LIM domain protein (LIM3), and PRELI domain-containing protein 2 (PRELI) in RS (Figure 3E). MAP7 corresponds to the gene encoding the ensconsin. LIM3 encodes an RNA polymerase II transcription factor with a key role in neuron specification. PRELI, a protein-coding gene, may be involved in phosphatidic acid transfer activity and phospholipid transport and is located in the mitochondrial intermembrane space. The protection against acetamiprid may be attained through the downregulation of these genes.

## 4. Discussion

In the present study, we have obtained a strain RS of pea aphid that exhibited reduced susceptibility to acetamiprid and characterized it by transcriptomic and metatranscriptomic approaches. Most intriguingly, the RS strain, despite being exposed to acetamiprid for nine consecutive generations, exhibited only a twofold increase in LC_50_ compared to the unselected SS strain. In fact, documented cases of insecticide resistance in pea aphids are rare. Very few cases have been reported so far against pyrethroids [34,51]. This rarity suggests that resistance to most insecticides is low in this species. Furthermore, we found that key players in cellular metabolism and stress response, such as TPI, PHI, and P5CS, were downregulated in the RS strain. This suggests that the insect has adopted strategies, such as metabolic reprogramming, that prioritize survival over growth to mitigate the negative effects of acetamiprid. While this adaptive response may improve short-term survival, it may impose a long-term fitness cost on the insect as resources are diverted from other essential biological functions, such as growth. Indeed, we observed differences in the growth cycle between the RS and SS strains: the RS strain required an additional 24 h to reach the adult stage, indicating a possible fitness cost in the absence of acetamiprid.

Both transcriptomic sequencing and qRT-PCR experiments have revealed the upregulation of ACSF2 and FAR, two important enzymes involved in lipid metabolism, in the RS strain. ACSF2 facilitates the conversion of fatty acids into fatty acyl-CoA, a crucial step in fatty acid metabolism. Meanwhile, FAR plays a key role in the conversion of fatty acyl-CoA into fatty alcohols, which are essential for the synthesis of cuticular hydrocarbons (CHCs). CHCs form the protective wax layer found on the surface of many aphids and mealybugs, shielding them from natural enemies and adverse environmental conditions [52,53]. The pea aphid is known to have a wax layer covering all parts of its body [54]. The constitutive upregulation of ACSF2 and FAR in RS due to prolonged exposure to acetamiprid may have increased the production of CHCs and, hence, the wax layer, ultimately reducing acetamiprid penetration. This explanation needs further investigation.

RNAi-mediated suppression of FAR expression in the cotton mealybug, *Phenacoccus solenopsis* Tinsley, resulted in reduced CHC levels in the wax layer [55]. Mealybugs with reduced CHC content exhibited increased mortality when exposed to desiccation and deltamethrin treatments, highlighting the importance of this enzyme for insect adaptation to water loss and insecticide stress. Therefore, we hypothesize that the potential mechanism responsible for reduced susceptibility to acetamiprid in the RS strain involves the thickening of the physical barrier through the wax layer that prevents the penetration of acetamiprid molecules into the aphid’s body. This process may be associated with changes in wax composition and layer deposition. Future work will characterize the wax content of the RS strain compared to the SS strain in terms of wax content by gravimetric analysis and wax components by gas chromatography–mass spectrometry (GC–MS). While an increased wax layer may confer resistance to insecticides, it may also impose fitness costs on the aphids, as the production of a thicker or more abundant wax layer requires resources and energy that could otherwise be used for essential biological processes such as growth.

Interestingly, genes encoding ribosomal proteins showed significant upregulation in the RS transcriptome over the SS transcriptome. While ribosomal proteins are primarily known for their involvement in protein synthesis within ribosomes, they have been found to have additional functions in various cellular processes. These proteins may interact with other molecules, such as proteins or nucleic acids, to exert these non-canonical functions [56]. Previously, Yu et al. [57] linked the ribosomal protein S29 to deltamethrin resistance by binding to CYP6N3—a member of the CYP6 class of cytochrome P450 enzymes involved in metabolic resistance to pyrethroids—and stimulating its degradation by the 26S proteasome. Overexpression of RPS29 reduced cell viability in the presence of deltamethrin. In addition, another study identified the ribosomal protein RpS2 as a potential receptor for the insecticidal protein Vip3Aa from *Bacillus thuringiensis* [58]. RNAi-mediated silencing of RpS2 gene expression in both transfected Sf21 cells and in larvae of *Spodoptera litura* (Fabricius, 1775) injected with double-stranded RNA resulted in reduced toxicity of the Vip3A protein. Further evidence for the specific upregulation of ribosomal proteins in response to a selective agent was provided by HT-SuperSAGE analysis of a Vip3Aa-selected population of *Heliothis virescens* (Fabricius, 1777), which confirmed this phenomenon [59]. Whether the constitutive overexpression of ribosomal proteins in acetamiprid-selected pea aphids responds to an increased demand for protein synthesis in response to stress or is involved in a specific adaptive mechanism remains to be elucidated.

In our study, we detected two secondary symbionts of pea aphids, namely *H. huttiense* and *D. acidovorans*, exclusively in RS, while *L. iners* was more prevalent in RS compared to SS. The precise impact of these secondary symbionts on the adaptive response of RS to acetamiprid remains uncertain. Some research suggests that symbiotic bacteria actively participate in detoxifying insecticides. For instance, *Serratia oryzae* has been implicated in insecticide resistance in *Aedes albopictus* (Skuse, 1894), contributing to resistance development by upregulating the expression and activity of metabolic detoxification enzymes in mosquitoes [60]. In *A. gossypii*, the composition of symbiotic bacteria undergoes significant changes after insecticide treatment [61]. Notably, antibiotic treatment has also been found to increase the sensitivity of *A. gossypii* to spirochetes [62]. However, the interaction between aphids and symbionts is intricate. While aphids may gain benefits from hosting symbionts, negative effects could lead to the reallocation of aphid energy resources [63]. For instance, infection of the corn leaf aphid, *Rhopalosiphum maidis* (Fitch, 1856), with *H. defensa* and *Regiella insecticola* could have a partially negative effect on aphid growth and development, although endosymbionts were maintained in aphids over time [64]. Previously, these two symbionts were shown to protect aphids, with *R. insecticola* shielding *A. pisum* from the aphid-specific fungal entomopathogen *Zoophthora occidentalis* [65], and *H. defensa* reducing aphid susceptibility to insecticides [31].

In summary, we have shown that acetamiprid-selected pea aphid RS strain exhibits changes in gene expression, the most interesting of which was the constitutive overexpression of genes related to the synthesis of the components of the epidermal wax layer. Overall, the development of an increased waxy layer represents a mechanism by which the pea aphid can mitigate the effects of acetamiprid. This emphasizes the importance of understanding wax biosynthesis and its role in acetamiprid resistance for effective pest management strategies. We have also shown the differential infection of the acetamiprid-selected RS strain with two specific secondary symbionts. Understanding the trade-offs that aphids must make to survive acetamiprid exposure while harboring these secondary symbionts is crucial for developing sustainable pest management strategies that consider both immediate efficacy and long-term ecological impact.

## Figures and Tables

**Figure 1 insects-15-00274-f001:**
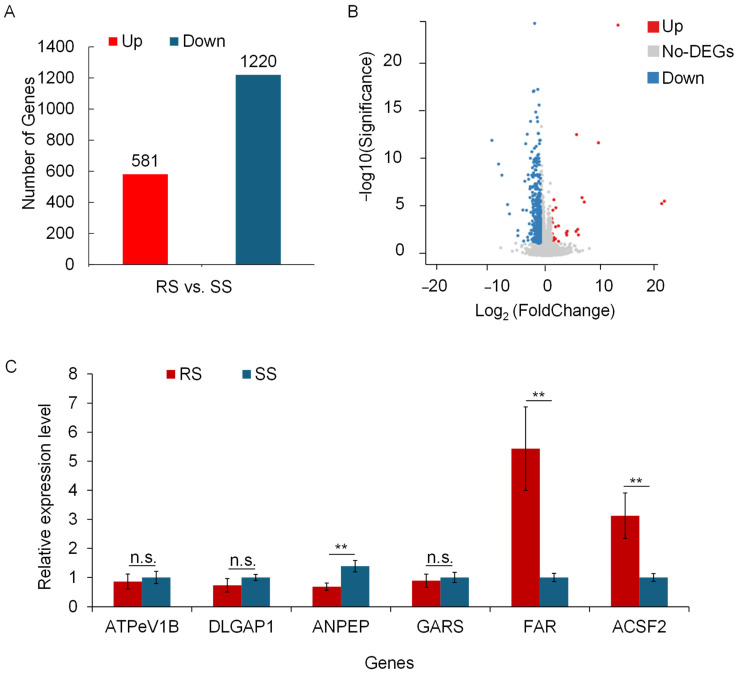
Analysis of differentially expressed genes (DEGs) between RS and SS aphid strains in the transcriptome. (**A**) Bar chart of the number of DEGs. Red represents the number of upregulated DEGs in RS, while blue represents the number of downregulated DEGs. (**B**) Volcano plot of significant DEGs. Dots represent individual genes. Blue dots represent significantly downregulated DEGs in RS, and red dots represent significantly upregulated DEGs. Grey dots indicate DEGs that are not significant between RS and SS. The data were analyzed with ANOVA (*p* < 0.05). (**C**) Quantitative real-time PCR (qRT-PCR) verification of ATP6V1, DLGAP1, FAR, and other genes after acetamiprid treatment. Each sample with different genes had three replicates. Significance analysis was conducted with ANOVA. Values are means ± SEM of three experiments. ** *p* < 0.01; n.s. not significant (*p* > 0.05).

**Figure 2 insects-15-00274-f002:**
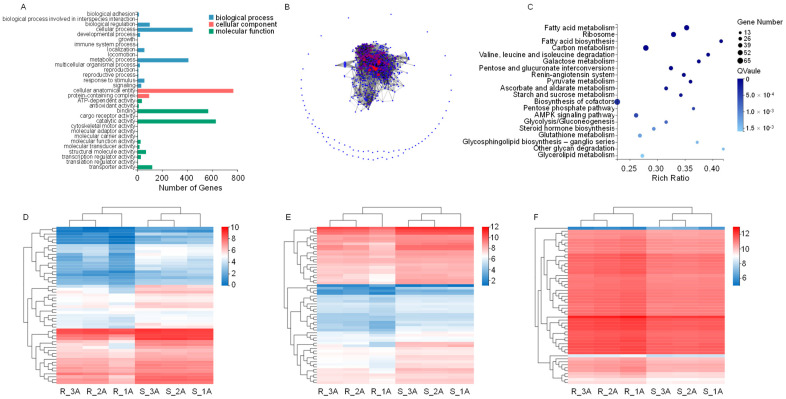
Gene ontology (GO) annotation and Kyoto Encyclopedia of Genes and Genomes (KEGG) pathway enrichment analysis of differentially expressed genes (DEGs) between RS and SS strains in the transcriptome. (**A**) GO classification of DEGs, showing the number of DEGs in different categories. Most of the DEGs were genes related to the biosynthesis of cellular anatomical entities, catalytic activity, and metabolic processes. (**B**) 628 genes of catalytic activity network with three genes at key nodes, including triosephosphate isomerase (TPI), glucose-6-phosphate isomerase (PHI) and delta-1-pyrroline-5-carboxylate synthetase (P5CS). (**C**) KEGG enrichment of DEGs. (**D**) Heatmap analysis of hierarchical clustering of DEGs in fatty acid metabolism. (**E**) Heatmap analysis of hierarchical clustering of DEGs in carbon metabolism. (**F**) Heatmap analysis of hierarchical clustering of DEGs in the ribosome. Red and blue indicate high and low expression in RS, respectively.

**Figure 3 insects-15-00274-f003:**
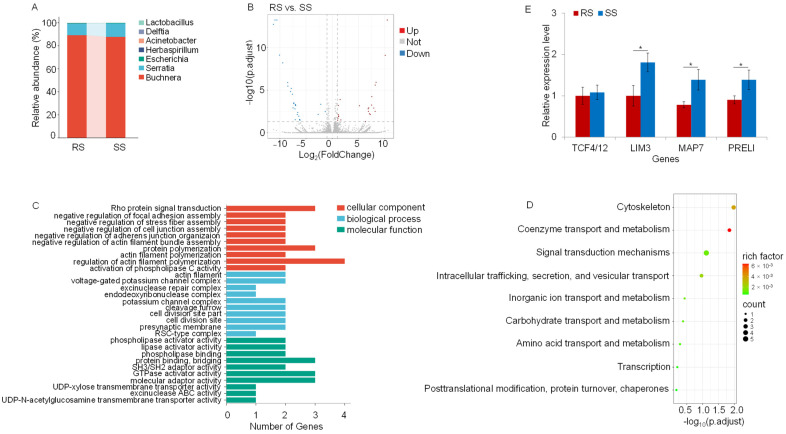
Analysis of differentially expressed genes (DEGs) between RS and SS strains in the metatranscriptome. (**A**) Bar chart of the relative abundance of bacteria. Different colors represent different species, according to the legend on the right. The color filling of the intervals and lines can visually reflect the changes in the relative abundance of the species. (**B**) Volcano plot of significant DEGs in the metatranscriptome. Dots represent individual genes. Blue dots represent significantly downregulated DEGs in RS, and red dots represent significantly upregulated DEGs. Grey dots indicate DEGs that are not significant between RS and SS. The data were analyzed using ANOVA (*p* < 0.05). (**C**) Gene ontology (GO) classification of downregulated DEGs in the metatranscriptome. (**D**) EuKaryotic Ortholog Groups (KOG) of downregulated DEGs. The horizontal axis represents the gene ratio. The bubble color indicates the *p* value, and the color gradient represents the magnitude of the *p* value, converted from −log_10_. The closer the color is to red, the lower the *p* value and the more significant the enrichment. The size of the bubbles indicates the number of DEGs in the functional class (the larger the bubbles, the greater the quantity). (**E**) Real-Time Quantitative Reverse Transcription PCR (qRT-PCR) of TCF4/12, LIM3, MAP7 and PREL. The qRT-PCR of the different genes was replicated three times. Significance analysis was conducted with ANOVA. Values are mean ± SEM of three experiments. * *p* < 0.05.

**Table 1 insects-15-00274-t001:** List of primers used in this study.

Genes	Forward Primer (5′−3′)	Reverse Primer (5′−3′)
ATP6V1 (NM_001293544)	TCGTCAAATCTATCCACCAA	AATGCCTCTTCTCCCACAAC
DLGAP1 (XM_029490798)	AATTCCTCGGTTTATGTGAG	ATTGCCTTGCGTTGTTCTTC
ANPEP (XM_001950011)	TTGGATGGGCATTGTTTCTA	ATAGTCCATATCACCGACCT
GARS (XM_003245009)	TCATTGCCTCCATTAGTAGC	ATTTGTTCCATTGAATCCCT
FAR (XM_003242260)	ACTACGAGTCACCACCTTTG	TTTCTGCTTTCGCATACATT
ACSF2 (XM_016806965)	CGCCAACTCTACAAGACAAC	CATGACAAGATACCCACGAA
TCF4/12 (XM_016808432)	TCGCCCGATGATGATAGTGT	GTGCCGTCCAAGTAATAAGA
LIM3 (XM_008188468)	GAACGCAGAACAGTAAAGAA	CTGGTATAATAACGGAGGAA
MAP7 (XM_008190631)	AGAGTTGCGGTTGCAGTTGG	TGTTGCTCGGCAGATTCAGT
PRELI (XM_029488204)	GAAGAATGTTGGTATGACGA	CATGTTGGATTTGGTGTAAT

**Table 2 insects-15-00274-t002:** Acetamiprid toxicity to non-selected SS and selected RS pea aphid strains expressed as LC_50_, LC_99_, and LC_5_. These values represent the concentration that causes death in 50%, 99%, and 5% of the population, respectively, expressed in µg acetamiprid per ml (ppm). In all cases, the 95% fiducial limits are given in parentheses.

Aphid Strain	LC_50_	LC_99_	LC_5_	Slope (± S.E.)	n ^a^	RR ^b^
SS	2.68 (2.46–2.92)	32.10 (31.34–32.87)	0.55 (0.01–1.09)	6.38 ± 0.37	150	–
RS	5.09 (4.88–5.31)	44.75 (44.02–45.48)	1.26 (0.81–1.72)	8.28 ± 0.56	150	1.9

^a^ Number of larvae used in the bioassays, including control. ^b^ RR, resistance ratio, is the LC_50_ for RS strain divided by the LC_50_ for SS strain.

**Table 3 insects-15-00274-t003:** Top 10 downregulated and upregulated DEGs in the RS transcriptome.

Gene ID	log_2_(R/S) *	Annotation
LOC115034459	−10.09	alpha-N-acetylgalactosaminidase
LOC100573163	−8.84	glycerol-3-phosphate dehydrogenase (NAD+)
LOC103308887	−8.23	SRA stem-loop-interacting RNA-binding protein, mitochondrial
LOC100168848	−7.17	NULL
LOC100572322	−6.82	polyhomeotic-like protein 1
LOC100570759	−5.26	MFS transporter, PAT family, solute carrier family 33 (acetyl-CoA transporter), member 1
LOC100570250	−5.26	lysophospholipid acyltransferase 7
LOC100159801	−4.33	gamma-glutamyltranspeptidase/glutathione hydrolase/leukotriene-C4 hydrolase
LOC103310139	−4.18	leucine-rich repeat and immunoglobulin-like domain-containing nogo receptor-interacting protein
LOC100162019	−3.96	lactase-phlorizin hydrolase
LOC100169462	13.39	V-type H+-transporting ATPase subunit B
LOC100575793	9.74	glycyl-tRNA synthetase
LOC100159545	7.11	discs, large-associated protein 1
LOC107882136	6.69	alanyl aminopeptidase
LOC100569077	6.02	NULL
LOC100571352	5.87	lysosomal acid phosphatase
LOC100570391	5.67	alcohol-forming fatty acyl-CoA reductase
LOC103310260	5.56	medium-chain acyl-CoA ligase, mitochondrial
LOC100570903	3.95	SWI/SNF related-matrix-associated actin-dependent regulator of chromatin subfamily C
LOC115034301	3.84	glutamate receptor, ionotropic, invertebrate

* log_2_(R/S) < 0: Downregulated DEG; log_2_(R/S) > 0: Upregulated DEG.

**Table 4 insects-15-00274-t004:** Abundance of bacteria.

Sample	*B. aphidicola* (%)	*S. symbiotica* (%)	*E. coli* (%)	*H. huttiense* (%)	*A. soli* (%)	*D. acidovorans* (%)	*L. iners* (%)
RS_1	87.97	11.09	0.44	0.32	0.18	0.0	0.0
RS_2	88.78	10.74	0.28	0.0	0.04	0.13	0.04
RS_3	90.93	8.07	0.18	0.02	0.01	0.0	0.79
SS_1	90.70	8.49	0.46	0.0	0.05	0.0	0.29
SS_2	86.42	13.34	0.24	0.0	0.0	0.0	0.0
SS_3	86.23	13.04	0.50	0.0	0.23	0.0	0.0

## Data Availability

The raw transcriptomic and metatranscriptomic sequencing data presented in this study are openly available in NCBI with SRA numbers SRR26347092 and SRR25998188, respectively.

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
