# Peer review of "Transcriptomic and Metatranscriptomic Analyses Provide New Insights into the Response of the Pea Aphid Acyrthosiphon pisum (Hemiptera: Aphididae) to Acetamiprid"

_insects, 2024, doi:10.3390/insects15040274_

Round 1
Reviewer 1 Report
Comments and Suggestions for Authors
Transcriptomic and metatranscriptomic analyzes provide new 2 insights into the response of the pea aphid Acyrthosiphon pisum 3 (Hemiptera: Aphididae) to acetamiprid is well designed study to show pea aphid adaptation to Acetamprid.Also add more implication of pest control in discussion. Few comments are mentioned in PDF file.

Author needs to improve Language especially in introduction
Author Response
Dear reviewer,
We would like to thank you very much for your kindly reading and comments on our manuscript to further improve its quality. We have made the changes listed below, which we believe meet the suggestions as far as possible.
Comments to the Author
C1: The author needs to improve Language especially in introduction
We thank the reviewer for this suggestion. In the revised manuscript, particular attention has been paid to revising the language in the introduction, including grammar, clarity, conciseness, and coherence, to improve readability and comprehension. The revised
C2: Line 56: non-target effects like human health, etc.
We thank the reviewer for this suggestion. The potential impact on human health and the environment of acetamiprid has been mentioned in the revised manuscript version (Lines 56-57)
C3: Lines 104-109: Not needed here.
We thank the reviewer for this suggestion. This part has been omitted in the revised manuscript version. We agree with the reviewer that this part belongs to the results and discussion, where it is already very detailed.
We believe that we have responded positively to the points raised by the reviewer, the revised manuscript is in the attachment.
Thanks again for your kindly work.

Reviewer 2 Report
Comments and Suggestions for Authors
The paper submitted by Cai et al. provided an analysis of transcriptomic and metatranscriptomic data obtained from acetamiprid-selected pea aphids compared to an unselected reference strain. The authors selected a pea aphid clone over nine consecutive generations with the common neonicotinoid insecticide acetamiprid by exposure to treated leaves. The resulting strain RS exhibited a two-fold tolerance to acetamiprid which was significant when compared to the unselected parental strain SS. RNAseq data and an analysis of differentially expressed genes provided insights into adaptive changes related to acetamiprid exposure. The authors generated interesting transcriptomic data which, deposited to GenBank, provide an interesting resource for future studies on pea aphids upon insecticide exposure. Expectedly more than 1700 genes were differentially expressed between strains. Their results indicated that particularly genes potentially involved in the formation of cuticular hydrocarbons were upregulated in acetamiprid selected aphids. Whereas the metatranscriptomic analysis provided weak differences between both strains, SS and RS.
The methods are well described and the paper is generally well referenced and written, albeit it remains quite speculative regarding the interpretation of its results, especially as the authors did not provide any experiments to functionally validate their claims concerning potential changes in the cuticle wax layer and its impact on acetamiprid penetration. I have a few, mostly minor points I want the authors to address.
1) Title: Please change “analyzes” to “analyses”
2) It is a pity the authors haven´t checked the wax content of the selected strain in comparison to the unselected strain. It is such a simple experiment: just washing the aphids with hexane, evaporating the solvent and gravimetric measurement of the remainder (lipids). Perhaps the authors can at least discuss the experiment (L390ff).
3) It should be mentioned that aphids exposed to leaves dipped into acetamiprid solution may also taking up the insecticide via ingestion as acetamiprid is known to be systemic.
4) The authors should avoid the term resistance, because a RR of 1.9 is indeed very low (e.g. 3.3).
5) Figure 2 needs to be enlarged because the fonts are too small.
6) The summary section is rater speculative, please adopt. It should be mentioned that the data will provide a valuable resource for future studies.
Author Response
Dear reviewer,
We would like to thank you very much for your kindly reading and comments on our manuscript to further improve its quality. We have made the changes listed below, which we believe meet the suggestions as far as possible.
Comments to the Author
C1: Title: Please change “analyzes” to “analyses”
This change has been made.
C2: It is a pity the authors haven´t checked the wax content of the selected strain in comparison to the unselected strain. It is such a simple experiment: just washing the aphids with hexane, evaporating the solvent and gravimetric measurement of the remainder (lipids). Perhaps the authors can at least discuss the experiment.
We thank the reviewer for this comment. Indeed, after analyzing the results of the current study, a new research project will aim to identify changes in the composition and quantity of wax in RS. Therefore, we mentioned in the "Summary" section that the results of this study will be relevant for future research projects. Nevertheless, in the revised manuscript we inform about our plans for future research to better characterize the wax composition and content in our group (lines 367-369).
C3: It should be mentioned that aphids exposed to leaves dipped into acetamiprid solution may also taking up the insecticide via ingestion as acetamiprid is known to be systemic.
We thank the reviewer for this comment. We have added a new sentence in the revised manuscript to include additional information regarding the systemic action of acetamiprid, highlighting its effectiveness in controlling target insects through both contact and ingestion (Lines 115-116).
C4: The authors should avoid the term resistance, because a RR of 1.9 is indeed very low (e.g. 3.3).
We thank the reviewer for this comment. In the revised manuscript, Subheadings 3.2 and 3.3, the term resistance was substituted by adaptation.
C5: Figure 2 needs to be enlarged because the fonts are too small.
We thank the reviewer for this comment. In the revised manuscript, Figure 2 was improved with a larger font size and high resolution to ensure clarity and readability.
C6: The summary section is rather speculative, please adopt. It should be mentioned that the data will provide a valuable resource for future studies.
We thank the reviewer for this comment. The speculative language from the Summary section has been removed and replaced with a more neutral and informative tone. The emphasis is on presenting the results as valuable insights into the adaptation process of pea aphids to acetamiprid exposure, without making assumptions about future effects or outcomes. The new text maintains the scientific integrity of the summary while emphasizing the importance of the research findings for future studies and pest management strategies.
The revised manuscript is in the attachment.
Thanks again for your kindly work.
Yours sincerely,
Dandan Li

Reviewer 3 Report
Comments and Suggestions for Authors
Reviewer Comment and Recommendation for Manuscript ID: Insects-2947461-peer-review-v1
Title: Transcriptomic and metatranscriptomic analyzes provide new insights into the response of the pea aphid Acyrthosiphon pisum (Hemiptera: Aphididae) to acetamiprid
General Comments: In general, the manuscript effectively addresses the topic. Nevertheless, there is room for additional edits to enhance readability and conciseness. I suggest the authors consider incorporating my specific comments and suggestions to facilitate the acceptance and publication of this manuscript in MDPI-Insects
Specific comments and suggestions:
Introduction
L.105-109: Move “Our results.........adaptation process.” to the results and discussion section. Also, make modification in L 108-109 to specify the objective about “analysis of microbiome of RS” strain of pea aphid in the introduction section. This objective should be in L.105 just after “expressed genes.”
Material and Methods
L.120-120. Replace “10 individuals each” with “10 individuals in each replication”
L.121-122: Add a sentence to explain why the F9 was used as the RS
L.195: Change “LC10” to “LC5”
Results
L.249-252: Move “The reason......pea aphid genome [49]” to the Discussion section
Discussion
L.369-370: Replace “Only one case has” with “Very few cases have” because the authors provided two citations “[34,50]
L.379: Should “absence of acetamiprid” be replaced by “presence of acetamiprid” for the conclusion to agree with the premises?
L.414: Edit the text “ Bacillus thuringiensis”; italicize
Author Response
Dear reviewer,
We would like to thank you very much for your kindly reading and comments on our manuscript to further improve its quality. We have made the changes listed below, which we believe meet the suggestions as far as possible.
Comments to the Author
C1: Introduction, L.105-109: Move “Our results.........adaptation process.” to the results and discussion section. Also, make modification in L 108-109 to specify the objective about “analysis of microbiome of RS” strain of pea aphid in the introduction section. This objective should be in L.105 just after “expressed genes.”
We thank the reviewer for this comment. The fragment referring to “our results… adaptation process” has been removed from the introduction. We agree with the reviewer that this part belongs to the results and discussion, where it is already very detailed. We have also slightly changed the wording of the last paragraph of the Introduction in the revised manuscript to emphasize that the investigation of constitutive changes in gene expression and the microbiome were the two aims of this study, as we say, “to identify possible target mechanisms and symbionts involved in the adaptation process”.
C2: Material and Methods, L.120-120. Replace “10 individuals each” with “10 individuals in each replication”
This change has been made.
C3: Material and Methods, L.121-122: Add a sentence to explain why the F9 was used as the RS.
We thank the reviewer for this comment. The choice of F9 as the RS strain was based on a specific criterion defined during the experimental design phase. This criterion referred to notable differences between the RS and non-selected (SS) strains in terms of the time required to complete the growth cycle in F9. We have now included a sentence in the manuscript mentioning this (lines 121-123).
C4: Material and Methods, L.195: Change “LC10” to “LC5”.
This change has been made.
C5: Results, L.249-252: Move “The reason......pea aphid genome [49]” to the Discussion section.
We agree with the reviewer's suggestion to relocate this sentence to the discussion section rather than the results. However, upon review, we determined that the information conveyed in this sentence would not contribute significantly to the discussion of the main findings of our study. Therefore, the entire sentence has been omitted from the revised manuscript.
C6: Discussion, L.369-370: Replace “Only one case has” with “Very few cases have” because the authors provided two citations “[34,50].
This change has been made.
C7: Discussion, L.379: Should “absence of acetamiprid” be replaced by “presence of acetamiprid” for the conclusion to agree with the premises?
We thank the reviewer for this comment. Actually, the accurate statement is "absence of acetamiprid," as we are referring to the procedure followed to obtain the RS strain, wherein surviving aphids were transferred to fresh leaves after 72 hours of exposure to acetamiprid and allowed to complete the growth cycle. Indeed, fitness costs associated with insecticide adaptation or resistance are often observed in the absence of insecticide pressure. In environments where insecticides are not present, resistant individuals may experience reduced fitness compared to susceptible individuals. This reduced fitness can result from genetic mutations or physiological adaptations that confer resistance but also impose trade-offs in other aspects of the insect's biology. Consequently, resistant individuals may be less competitive in natural environments, leading to a decrease in their overall population fitness over time.
C8: Discussion, L.414: Edit the text “ Bacillus thuringiensis”; italicize.
This change has been made.
The revised manuscript is in the attachment.
Thanks again for your kindly work.
Yours sincerely,
Dandan Li
